# Are Coping Strategies, Emotional Abilities, and Resilience Predictors of Well-Being? Comparison of Linear and Non-Linear Methodologies

**DOI:** 10.3390/ijerph19127478

**Published:** 2022-06-18

**Authors:** Laura Lacomba-Trejo, Joaquín Mateu-Mollá, Monica D. Bellegarde-Nunes, Iraida Delhom

**Affiliations:** 1Departament of Personality, Evaluation and Psychological Treatment, Faculty of Psychology, University of Valencia, Av. Blasco Ibánez, 21, 46010 Valencia, Spain; laura.lacomba@uv.es; 2Faculty of Health Sciences, Valencian International University, Street Pintor Sorolla, 21, 46002 Valencia, Spain; jmateu@universidadviu.com; 3Fundación Pilares para la Autonomía Personal, Street de Ríos Rosas, 11, 28003 Madrid, Spain; monica.bellegarde@fundacionpilares.org; 4Departament of Developmental, Educational and Social Psychology and Methodology, Universitat Jaume I Castellón, Av. Vicent Sos Baynat, s/n, 12071 Castellón de la Plana, Spain

**Keywords:** emotional intelligence, problem-oriented coping, resilience, satisfaction with life, positive and negative affect, prediction

## Abstract

Emotional intelligence (EI), problem-oriented coping, and resilience have been deeply studied as psychological predictors of wellbeing in stressful daily situations. The aim was to find out whether coping, EI, and resilience are predictors of well-being, using two statistical methodologies (hierarchical regression models and comparative qualitative models). With this objective in mind, we built an online evaluation protocol and administered it to 427 Spanish people, exploring these variables through a selection of validated tests. The extracted data were studied using linear predictive tests (hierarchical regression models), as well as fuzzy set qualitative comparative analysis. We found that EI variables had important associations with coping, positive affect, negative affect, and life satisfaction, and also acted as relevant predictors for all of them, together with resilience and problem-oriented coping. The fuzzy set qualitative comparative analysis showed a series of logical combinations of conditional causes and results of each potential configuration for these variables. The interaction between the presence of EI, resilience, and coping resulted in high levels of well-being. On the other hand, the presence of high emotional attention in interaction with low resilience and low coping abilities resulted in low well-being. These results increase knowledge about protective factors and allow for the creation of intervention programmes to enhance them.

## 1. Introduction

In recent years, Positive Psychology has gained ground in the traditional study of psychology, moving away from an exclusive focus on psychopathology, and offering a complementary salutogenic approach based on the knowledge of people’s positive qualities [1]. From this perspective, the conception of health is broad, being considered a state of physical, mental, and social well-being, and not simply the absence of illness, thus highlighting the concept of well-being [2].

In psychology, well-being is conceptualised in terms of subjective and psychological well-being. Subjective well-being has been defined as the cognitive and affective evaluation of one’s own life. This construct comprises two dimensions: the cognitive dimension, which refers to life satisfaction; and the affective dimension, consisting of positive and negative affect [3]. Positive affect involves experiencing pleasant moods and emotions, whereas negative affect involves unpleasant emotions and moods [4]. On the other hand, life satisfaction refers to people’s overall evaluation of themselves, and the degree to which they experience a sense of well-being [5]. It is based on beliefs and attitudes about one’s life, and is a significant indicator of positive personal, psychological, social, interpersonal, and intrapersonal outcomes [6]. Given the importance of well-being in people’s lives, it is of great value to know which factors may be predictors of well-being in order to show the value of predictive models of well-being, and to contribute to the design of interventions that work to improve it.

In this regard, one of the constructs that have been related to well-being is emotional intelligence (EI) [6,7,8]. EI is defined as the ability to perceive, evaluate, and express one’s emotions accurately, the ability to access and generate feelings that facilitate thought, the ability to understand emotion and emotional knowledge, and the ability to regulate emotions and promote emotional and intellectual growth [9]. Specifically, these emotional abilities are configured through three dimensions: attention, abilities clarity, and emotional repair [9,10]. EI involves a set of emotional abilities to effectively use the information provided by emotions, which allows the application of more adaptive behavioural and cognitive repertoires when coping with stressful situations [11].

EI abilities are closely linked to coping, which is understood as those cognitive and behavioural efforts used to deal with stressful or conflict situations [12]. EI and successful coping have been mainly related to problem-focused coping strategies, such as planning, positive re-appraisal, and seeking social support [4,13,14]. Furthermore, both EI and adaptive coping strategies appear to be variables closely linked to psychological resilience and have been found to work together to predict different positive mental health outcomes [11,15,16]. Resilience is understood as the ability to persist, grow, be strong, and even succeed in life, despite adversity; it enables people to successfully recover when faced with an obstacle, and to move forward strengthened despite difficulties [17].

EI, problem-focused coping strategies, and resilience are key to adaptation processes [4,18,19]. These constructs facilitate appropriate responses in stressful situations, and decrease maladaptive emotional reactions, promoting positive moods and reducing the negative ones [4,20]. In this regard, Limonero and colleagues [21] found that people with high levels of EI are more likely to show high levels of resilience and use more adaptive coping strategies that help them reduce stress and increase life satisfaction. Similarly, Ramírez-Fernández and colleagues [22] found a significant positive relationship between resilience and life satisfaction and positive affect. Along the same lines, Mateo and colleagues [14] found that problem-focused strategies were predictive of life satisfaction and positive affect. However, these studies addressed the role of IE, problem-focused strategies, and resilience on well-being through linear methodologies. We consider that knowing the specific role that each of these variables plays on well-being in its different forms is relevant for the definition of more effective intervention programmes. In addition to the above, these studies have not been carried out in the Spanish population. Through our work, we are able to expand our knowledge of the behaviour of these variables in the Spanish context.

This study aims to find out whether EI abilities, problem-focused coping strategies, and resilience are predictors of subjective well-being in the general population. To this end, linear and non-linear methodologies are used in their prediction. This combination of methodologies allows studying the relationship between the analysed variables in much greater depth. In addition, QCA models allow the observation of the different paths or combinations that lead to the same result, providing great value to the present study.

## 2. Methods

### 2.1. Participants

The evaluation protocol was completed by 427 Spanish participants (78.50% women), aged between 18 and 83 years (M = 35.81; SD = 11.18): 33.50% of them were single, 28.60% were married, 4.70% were divorced, and the last 0.70% were widowers. Concerning the employment situation, 60.40% worked for others, 10.50% worked as freelancers, 23.70% were unemployed, 1.60% were retired, and 1.90% were students. In terms of academic level, 82.40% of the sample had a university education, 15.80% had secondary studies, and 1.60% had primary studies. All the participants: (a) were persons of Spanish nationality, (b) completed the questionnaires correctly (responded to the battery completely or responded non-randomly), (c) signed the informed consent, and (d) did not suffer from severe physical diseases or mental disorders (personality disorders and neurological or oncological pathologies).

### 2.2. Measures

Sociodemographic variables: to analyse the study variables, an ad hoc questionnaire was designed that explored: sex, age, civil status, academic level, employment situation, and physical/mental health history.

Emotional Intelligence: Trait Meta Mood Scale–24 (TMMS-24) [23], adapted to Spanish [10]. The test consists of 24 Likert items with options according to the level of agreement (from 1 “absolutely not agree” to 5 “totally agree”). It has a tri-factorial structure, including questions about how people attend to their emotions (emotional attention), how they identify them (emotional clarity), and how they solve them (emotional reparation), with eight items for each one. The test has good psychometric properties [10]. In this study, the internal consistency was α = 0.88 for attention, α = 0.88 for clarity, and α = 0.86 for reparation.

Stress-coping: Coping with Stress Questionnaire (CSQ) [24]. This is a self-report measure with 42 items about people’s different coping strategies for demanding situations. The items have a Likert structure with five options, from 0 (“never”) to 4 (“always”). All the factors related to problem-focused coping were selected: seeking social support, problem-solving, and positive re-evaluation. The psychometric properties were good for all in previous studies [25]. In this investigation, adequate reliability indices were found for seeking social support (α = 0.94), problem-solving (α = 0.86), and positive re-evaluation (α = 0.79).

Resilience: Brief Resilience Coping Scale (BRCS) [26], adapted to Spanish [27]. The BRCS comprises four items with a Likert structure and a response spectrum from 1 to 5, according to the level of agreement. Its reliability levels are good [27]. In this study, α = 0.69 with Cronbach’s alpha was found.

Satisfaction with life: Satisfaction with Life Scale (SWLS) [28], adapted to Spanish [29]. The SWLS allows determining the subjective perception about self-existence, providing five items with seven response options. The test has shown excellent psychometric properties [29]. For this study, the reliability was α = 0.90.

Positive and negative affect: Positive and Negative Affect Schedule (PANAS) [30], adapted to Spanish [31]. The PANAS has a bi-dimensional structure, made of two factors with the same number of items (10 each). Its reliability levels and temporal consistency suggest that it might be used as a personality evaluation test [32]. The reliability levels were good in this study, with α = 0.91 for negative affect and α = 0.93 for positive affect.

### 2.3. Procedure

After signing the informed consent, the participants accessed the battery through Google Forms. This battery was designed specifically for the current study. In total, 20 min were required to complete the protocol. This evaluation strategy was chosen due to the COVID-19 health crisis restrictions. The methodology guidelines were developed according to the informed and deontological requirements from the Helsinki Declaration [33]. In addition, the study was approved by the ethics committee of the Valencian International University (CEID2021_15).

### 2.4. Data Analysis

In the first place, descriptive statistics were conducted for the total sample; including central tendency (mean), dispersion (standard deviation), position (percentiles), and rank measures (minimum and maximum) for all the dependent variables. In order to determine the covariations for preparing predictive analysis, Pearson’s correlations and hierarchical regressions in three steps were used. All these analyses were performed by using the 26th version of SPSS.

Finally, a fuzzy-set qualitative comparative analysis was conducted by using the fsQCA software [34]. For this, raw data were transformed into fuzzy sets, deleting lost data and recalibrating all the constructs [35]. After that, necessary and sufficient tests were run for testing the EI, active coping, and resiliency effects upon life satisfaction, positive affect, and negative affect. With the purpose of identifying the necessary conditions, fsQCA analysis performs an algorithm that transforms all data into a truth table that displays all logical combinations of conditional causes and results of each potential configuration. Ultimately, the software generates three possible solutions: complex, parsimonious, and intermediate. The latter is the most recommended, and the one selected for this study [36]. The sufficiency analysis considers that coverage solution refers to explained variance (number of observations that can be explained by a specific combination of conditions), whereas the consistency solution refers to the model reliability.

## 3. Results

### 3.1. Descriptive Statistics and Relationship between Variables

#### 3.1.1. Descriptive Statistics

Moderate punctuations were observed for emotional attention (M = 30.01; SD = 5.93), clarity (M = 29.96; SD = 5.59), and repair (M = 29.19; SD = 5.92). Similar results were obtained for the problem-focused coping dimensions: seeking social support (M = 14.40; SD = 6.75), problem-solving (M = 16.23; SD = 4.69), and positive re-evaluation (M = 16.41; SD = 4.27). Resilience reached middle-high levels (M = 14.62; SD = 3.09). Satisfaction with life obtained moderate scores (M = 23.03; SD = 7.10), whereas positive affect achieved high levels (M = 35.59; SD = 8.67). Negative affect was moderate (M = 21.86; SD = 8.94) (Table 1).

#### 3.1.2. Correlational Analysis

The correlational analysis showed that EI factors were associated with all the dependent variables included in the investigation. Emotional attention had negative correlations with resilience (r = −0.11; *p* < 0.05), positive affect (r = −0.14; *p* < 0.01), and life satisfaction (r = −0.11; *p* < 0.05), in addition to a direct association with negative affect (r = 0.35; *p* < 0.05). Emotional clarity and reparation had positive and significant correlations with all the dependent variables (*p* < 0.001), with the exception of negative affect (r = −0.28; *p* < 0.001 and r = −0.32; *p* < 0.001, respectively). Similar results were also found in coping strategies, significantly correlated with all the dependent variables, except negative affect. In this case, positive re-evaluation (r = −0.26; *p* < 0.001) and problem-solving (r = −0.22; *p* < 0.001) were inversely associated with it. Satisfaction with life obtained positive correlations with all the dependent variables, except for emotional attention and negative affect, which were negative. Further information can be found in Table 2.

### 3.2. Hierarchical Regression Models vs. QCA

#### Hierarchical Regression Models

The predictive power of the variables was analysed through a hierarchical regression model. Satisfaction with life, positive affect, and negative affect were considered as criterion variables, whereas EI, problem-focused coping, and resilience were emplaced as the predictive dimensions. Three steps were established in the model: in the first place, the EI variables were included, followed by the protective factors (problem-focused coping) in the second step, and resilience in the last step.

In the first step, the EI variables significantly increased the variance of satisfaction with life (Δ*R*^2^ = 0.29, *p* ≤ 0.001), positive affect (Δ*R*^2^ = 0.33, *p* ≤ 0.001), and negative affect (Δ*R*^2^ = 0.25, *p* ≤ 0.001). In the second step, where the problem-focused coping was incorporated, the variance of satisfaction with life (Δ*R*^2^ = 0.08, *p* ≤ 0.001) and positive affect (Δ*R*^2^ = 0.09, *p* ≤ 0.001) grew moderately, unlike the negative affect (Δ*R*^2^ = 0.00, *p* > 0.05). In the third and last step, just the positive affect variance increased (Δ*R*^2^ = 0.01, *p* ≤ 0.01).

Regarding the third step (which includes all the model variables), significant standard beta coefficients were found in emotional attention (β = −0.13; *p* ≤ 0.001), emotional clarity (β = 0.10; *p* ≤ 0.05), emotional reparation (β = 0.28; *p* ≤ 0.001), seeking social support (β = 0.21; *p* ≤ 0.001), and positive re-evaluation (β = 0.13; *p* ≤ 0.05). Regarding positive affect, it was obtained in emotional attention (β = −.13; *p* ≤ 0.001), emotional clarity (β = 0.13; *p* ≤ 0.01), emotional reparation (β = 0.16; *p* ≤ 0.001), positive re-evaluation (β = 0.27; *p* ≤ 0.001), and resilience (β = 0.14; *p* ≤ 0.01). Lastly, regarding negative affect, significant standard beta coefficients were found in emotional attention (β = 0.34; *p* ≤ 0.001), emotional clarity (β = −0.21; *p* ≤ 0.001), and emotional regulation (β = −0.15; *p* ≤ 0.05). Altogether, the final model explained 36% of the variance of satisfaction with life, 42% of the variance of positive affect, and 25% of the variance of negative affect (Table 3).

### 3.3. Fuzzy Set Qualitative Comparative Fuzzy Set Analysis (fsQCA)

#### 3.3.1. Analysis of Necessity

First, the main descriptors and calibration values for the study variables are presented (Table 4). Based on the results obtained, there were no necessary conditions for the high and low levels of life satisfaction, and positive and negative affect, as the consistency was lower than 0.90 in all cases [36].

#### 3.3.2. Analysis of Sufficiency

In reference to the sufficiency analysis, the combination of conditions that led to high and low levels of life satisfaction, and positive and negative affect were calculated (Table 5). Based on the premise that in fsQCA, a model is informative when the consistency is around or above 0.74 [37], all models obtained were consistent.

In the prediction of high levels of life satisfaction, twelve pathways explained 58.00% of the high levels (overall consistency = 0.82; overall coverage = 0.58). The most relevant pathway was the interaction between high levels of resilience, problem-solving, positive re-evaluation, and social support (Raw coverage = 0.36; Consistency = 0.88). In the prediction of low levels of life satisfaction, sixteen combinations explained 78.00% of low levels (overall consistency = 0.77; overall coverage = 0.78). The most relevant pathway that explained low levels of life satisfaction was the combination of the interaction between low levels of positive re-evaluation, repair, and emotional clarity, explaining 56.00% of cases (raw coverage = 0.56; consistency = 0.85).

Regarding the analyses conducted for the dependent variable positive affect, the intermediate solution indicated that ten combinations of causality explained high levels of positive affect and accounted for 80.00% of cases (overall consistency = 0.80; overall coverage = 0.65). In this prediction, the most relevant pathway was the result of the interaction between the presence of problem-solving, repair, and emotional clarity (raw coverage = 0.48; consistency = 0.85). On the other hand, in the prediction of low levels of positive affect, fifteen pathways were observed that explained 88.00% of the cases with low levels (overall consistency = 0.75; overall coverage = 0.88). The most relevant pathway to predict low levels of positive affect resulted from the combination of the absence of emotional repair and positive re-evaluation, explaining 65.00% of cases (raw coverage = 0.65; consistency = 0.88).

In the prediction of high levels of negative affect, three pathways explained 31.00% of the high levels (overall consistency = 0.80; overall coverage = 0.31). The most relevant pathway was low levels of emotional repair and problem-solving, in interaction with high levels of emotional attention, positive re-evaluation, and social support (raw coverage = 0.22; consistency = 0.84). In low levels of negative affect, thirteen pathways explained 53.00% of the high levels (overall consistency = 0.91; overall coverage = 0.53). The most relevant pathway was the interaction between high levels of emotional repair and clarity and problem-solving with low levels of emotional attention (raw coverage = 0.25; consistency = 0.94).

## 4. Discussion

The study of protective psychological factors that contribute to well-being is currently of great interest, especially variables such as EI, coping, and resilience [4,11,14,20,38]. However, few studies examine the predictive value of these variables on well-being. Moreover, the studies that provide evidence on their predictive value do so exclusively using linear methodologies. This study provides evidence of the predictive power of EI dimensions, problem-focused coping, and resilience on well-being (positive affect, negative affect, and satisfaction), using a combination of linear and non-linear methodologies that allow to compare both results and further examine the relationships between variables.

The results obtained through HRM show that the three EI dimensions (attention, clarity, and repair), positive re-appraisal strategies, and the search for social support are explanatory of life satisfaction. Moreover, EI dimensions, positive re-appraisal strategy, and resilience all predict positive affect, whereas negative affect is predicted exclusively by EI dimensions. These results suggest that EI may be particularly relevant in predicting well-being in both its affective and cognitive components.

Although all independent variables were found to be predictors of well-being, EI dimensions were particularly relevant. Previous studies have shown the relevance of EI in relation to well-being [7,8,20], considering that this set of abilities not only helps people to cope effectively with unpleasant emotions, but also promotes positive emotions that foster personal growth and well-being [20]. In addition, EI is considered to facilitate emotional recovery, reducing the initial impact of stressful or conflict situations [39].

In relation to the results obtained through QCA, high levels of resilience, problem-solving, positive re-appraisal, and seeking social support explained the high levels of satisfaction. These results suggest that satisfaction is particularly dependent on problem-focused coping and resilience. These abilities involve setting goals, planning to achieve them, and redirecting attention to positive aspects of life. Mateo and colleagues [14] argue that these virtues provide a sense of coherence and give meaning to one’s life, and can, therefore, influence people’s levels of well-being. Low levels of life satisfaction, on the other hand, arise from the interaction between low levels of positive re-appraisal, repair, and emotional clarity. Thus, although one of the three pathways that most explained high levels of satisfaction included the presence of EI, the results suggest that the absence of emotional abilities is more relevant for the prediction of low levels of satisfaction than the presence of EI for high levels of satisfaction. Positive re-appraisal emerges as a key strategy for predicting high and low life satisfaction. Previous studies have emphasised this strategy as an important explanation when considering well-being, which supports the results obtained in this study [4,14].

High positive affect was predicted by problem-solving, emotional repair, and emotional clarity, whereas low levels of positive affect were explained by the absence of emotional repair and positive re-appraisal. Previous studies have linked emotional clarity and repair to problem-solving strategy, supporting the interaction between these variables in explaining affective states [13]. However, positive re-appraisal is also commonly shown to be related to these dimensions of EI and affect [13,14], contrary to the findings in this study with respect to high levels of positive affect. The potential of the analyses used allows for further exploration of the role of specific strategies, suggesting that the absence of positive re-appraisal of situations is even more relevant than its presence in the prediction of well-being. In relation to emotional repair, [40] tested a predictive model of EI on well-being, in which emotional repair explained positive affect to a greater extent than the rest of the EI dimensions, coinciding with the results obtained in this study. For negative affect, high levels were predicted by the absence of emotional repair and problem-solving in interaction with high levels of mindfulness, positive re-appraisal, and seeking social support. On the other hand, low levels of negative affect were explained by high levels of repair, clarity, and problem-solving in interaction with low levels of attention to emotions. Therefore, considering these results, it can be concluded that attention to emotions is a determinant in the occurrence of negative affect [8,13]; in fact, this dimension of EI is the most related to negative affective states [41]. People with moderate attention to emotions show good intrapersonal functioning, but high levels of attention are related to ruminative thoughts and to the intensification of emotions in unhealthy ways. Low levels of attention are associated with avoidant behaviours and negative mood states [8]. Likewise, the other dimensions of EI are shown to be determinant together with problem-focused coping strategies [20,38].

## 5. Conclusions

The results show how EI, problem-focused coping strategies, and resilience influence emotional and cognitive well-being. This suggests that these variables may be key elements in promoting well-being, as previous studies have shown [14,18,19]. The dimensions of emotional clarity and emotional repair intervene in positive affect and negative affect, whereas emotional attention influences negative affect. Coping strategies of problem-solving and re-appraisal are shown to explain well-being as a whole (satisfaction, positive affect, and negative affect), whereas seeking social support influences satisfaction and negative affect more than positive affect. These results, achieved through non-linear methodologies, show how combinations of variables play an important role in the outcome. This allows us to know that to contribute to positive affect and life satisfaction, we should focus on working on emotional repair, clarity, and coping strategies of problem-solving and re-appraisal, whereas to intervene on negative affect, social support is more relevant.

Finally, resilience seems to be particularly relevant in relation to life satisfaction and does not appear to have as much impact on current emotionality. Thus, the findings of the present study show resilience as a key variable to intervene in life satisfaction, as it seems to have a particularly significant impact on the subjective appraisal of one’s own life. However, although it also seems to contribute to current emotionality, it does so to a lesser extent.

Based on the valuable information provided by the statistical analyses applied, it is worth noting the added value of combining linear and non-linear methodologies to explore the relationships between these variables. This study provides enlightening results that allow understanding the specific role of each of the psychological strengths studied in relation to well-being, as well as the value of predictive welfare models. Moreover, it offers precise knowledge about the role of each of these variables for their application in the design of interventions aimed at improving people’s well-being. Such results need to be taken into consideration in the design of public policies that help to improve people’s well-being. Knowing the concrete way in which different variables predict well-being can contribute to more precise and effective designs for this purpose. However, this study is not without limitations. The sample, although adequate for the analyses performed, needs to be expanded. Moreover, it would be interesting to be able to look at these results in terms of socio-demographic variables. In addition, the majority of participants in the study were women and people with higher education. Future research needs a more equal gender ratio, as well as a more representative participation of people with no and little education; the aim is to check whether the same results are obtained and whether they can be generalised to the general population. This could increase the internal and external validity of the study. On the other hand, future studies should consider high-involvement management at work, as there is evidence that it plays an important role in life satisfaction [42]. Finally, it would be interesting to examine the role of socio-demographic variables, such as gender and educational attainment, in predicting well-being.

## Figures and Tables

**Table 1 ijerph-19-07478-t001:** Descriptive analysis (N = 427).

	TMMS-24				CAE		BRCS	PANAS		SWLS
	Emotional Attention (EA)	Emotional Clarity (EC)	Emotional Reparation (ER)	Seeking for Soc Support (SS)	Positive Re-Evaluation (PR)	Problem Solving (PS)	Resilience (RS)	Positive Affect (PA)	Negative Affect (NA)	Satisfaction with Life (SL)
M	30.01	29.96	29.19	14.4	16.41	16.23	14.62	35.59	21.86	23.03
SD	5.93	5.59	5.92	6.75	4.27	4.69	3.09	8.67	8.94	7.1
Min.	11	9	12	0	2	4	4	10	10	5
Max.	40	40	40	24	24	24	20	50	48	35
P10	22	23	21	4.8	11	10	11	23.8	11	11.8
P50	31	30	30	16	17	16	15	37	21	24
P90	38	38	37	23	22	22	19	46	34	31

**Table 2 ijerph-19-07478-t002:** Correlational analysis (N = 427).

	EA	EC	ER	SS	PR	PS	RS	PA	NA	SL
EA	/									
EC	0.08	/								
ER	−0.05	0.42 ***	/							
SS	0.19	0.28 ***	0.23 ***	/						
PR	−0.05	0.34 ***	0.61 ***	0.40 ***	/					
PS	−0.02	0.47 ***	0.50 ***	0.38 ***	0.49 ***	/				
RS	−0.11 *	0.45 ***	0.55 ***	0.23 ***	0.52 ***	0.63 ***	/			
PA	−0.14 **	0.40 ***	0.52 ***	0.29 ***	0.56 ***	0.47 ***	0.51 ***	/		
NA	0.35 ***	−0.28 ***	−0.32 ***	−0.02	−0.26 ***	−0.22 ***	−0.30 ***	−0.45 ***	/	
SL	−0.11 *	0.35 ***	0.51 ***	0.36 ***	0.48 ***	0.41 ***	0.41 ***	0.58 ***	−0.48 ***	/

Note: Emotional attention (EA), emotional clarity (EC), emotional reparation (ER), seeking social support (SS), positive re-evaluation (PR), problem-solving (PS), resilience (RS), positive affect (PA), negative affect (NA), and satisfaction with life (SL), * *p* ≤ 0.05; ** *p* ≤ 0.01 *** *p* ≤ 0.001.

**Table 3 ijerph-19-07478-t003:** Hierarchical regression (positive affect, negative affect, and satisfaction with life as the dependent variables).

**Predictor**	**Positive Affect**			
	**Δ*R*^2^**	**Δ*F***	**β**	** *t* **
Step 1	0.33	69.10 ***		
Emotional Attention			−0.14	−3.48 ***
Emotional Clarity			0.24	5.40 ***
Emotional Reparation			0.41	9.35 ***
Step 2	0.09	22.41 ***		
Emotional Attention			−0.14	−3.61 ***
Emotional Clarity			0.15	3.50 ***
Emotional Reparation			0.19	3.79 ***
Social Support			0.06	1.46
Positive Re-evaluation			0.29	5.73 ***
Problem-Solving			0.13	2.73 **
Step 3	0.01	7.30 **		
Emotional Attention			−0.13	−3.29 ***
Emotional Clarity			0.13	2.99 **
Emotional Reparation			0.16	3.21 ***
Social Support			0.07	1.67
Positive Re-evaluation			0.27	5.21 ***
Problem-Solving			0.08	1.45
Resilience			0.14	2.70 **
Durbin–Watson	1.69			
*R* ^2^ _ajd_	0.42 ***			
**Predictor**	**Negative Affect**			
	**Δ*R*^2^**	**Δ*F***	**β**	** *t* **
Step 1	0.25	47.41 ***		
Emotional Attention			0.36	8.44 ***
Emotional Clarity			−0.22	−4.72 ***
Emotional Reparation			−0.21	−4.42 ***
Step 2	0	0.78		
Emotional Attention			0.35	7.97 ***
Emotional Clarity			−0.22	−4.36 ***
Emotional Reparation			−0.16	−2.86 **
Social Support			0.04	0.83
Positive Re-evaluation			−0.08	−1.43
Problem-Solving			−0.01	−0.15
Step 3	0	1.39		
Emotional Attention			0.34	7.78 ***
Emotional Clarity			−0.21	−4.08 **
Emotional Reparation			−0.15	−2.57 *
Social Support			0.04	0.74
Positive Re-evaluation			−0.07	−1.2
Problem-Solving			0.02	0.33
Resilience			−0.07	−1.18
Durbin–Watson	1.87			
*R* ^2^ _ajd_	0.25 ***			
**Predictor**	**Satisfaction with life**			
	**Δ*R*^2^**	**Δ*F***	**β**	** *t* **
Step 1	0.29	58.16 ***		
Emotional Attention			−0.11	−2.57 *
Emotional Clarity			0.18	4.04 ***
Emotional Reparation			0.43	9.39 ***
Step 2	0.08	16.60 ***		
Emotional Attention			−0.14	−3.44 ***
Emotional Clarity			0.11	2.37 *
Emotional Reparation			0.29	5.54 ***
Social Support			0.21	4.58 ***
Positive Re-evaluation			0.14	2.70 **
Problem-Solving			0.06	1.18
Step 3	0	1.12		
Emotional Attention			−0.13	−3.29 ***
Emotional Clarity			0.1	2.14 *
Emotional Reparation			0.28	5.22 ***
Social Support			0.21	4.65 ***
Positive Re-evaluation			0.13	2.47 *
Problem-Solving			0.04	0.66
Resilience			0.06	1.06
Durbin–Watson	1.81			
*R* ^2^ _ajd_	0.36 ***			

Note: Δ*R*^2^ = Change on *R*^2^; Δ*F* = Change on *F*; β = regression coefficient; *t* = *t* value; * *p* < 0.05; ** *p* ≤ 0.01; *** *p* ≤ 0.01.

**Table 4 ijerph-19-07478-t004:** Main descriptions and calibration values.

	TMMS-24			CAE				SWLS	PANAS	
	Emotional Attention	Emotional Clarity	Emotional Repair	Social Support	Positive Re-Evaluation	Problem-Solving	Resilience	Life Satisfaction	Positive Affect	Negative Affect
M	75,069.74	76,453.13	63,754.11	3888.73	3816.39	4061.4	207.19	3647.37	1362,709.05	110,255.11
SD	93,564.26	101,222.27	84,319.61	4820.78	3813.83	4314.44	162.52	3857.08	2334,826.04	522,730.57
Min	8	2	8	1	3	8	1	1	1	1
Max	390,625	390,625	3,390,625	15,625	15,625	15,625	625	16,807	9,765,625	6,250,000
Calibration values										
P10	1728	3072	1440	16	316	192	36	48	2380	2
P50	38,400	36,864	27,648	1728	2880	2304	180	2250	345,600	384
P90	234,375	250,000	200,000	12,500	10,000	10,000	500	9072	4,000,000	112,473.6

Note: M; mean; SD: standard deviation; min: minimum; max: maximum; P10 = 10th percentile; P50 = 50th percentile; P90: 90th percentile.

**Table 5 ijerph-19-07478-t005:** Summary of the main sufficient conditions for the intermediate solution of life satisfaction, and positive and negative affect.

*Frequency* *Cut-Off: 1;*	High Levels of Life Satisfaction *Consistency Cut-Off: 0.92*			Low Levels of Life Satisfaction *Consistency Cut-Off: 0.92*			High Levels of Positive Affect *Consistency Cut-Off: 0.83*			Low Levels of Positive Affect *Consistency Cut-Off: 0.94*			High Levels of Negative Affect *Consistency Cut-Off: 0.85*			Low Levels of Negative Affect *Consistency Cut-Off: 0.92*		
	1	2	3	1	2	3	1	2	3	1	2	3	1	2	3	1	2	3
Emotional attention								○					●	●	●	○	○	○
Emotional clarity		●	●	○	○		●		●					○	○	●	●	
Emotional repair		●		○			●			○	○	○	○	○		●		
Social Support	●	●	●						●				●		●		○	
Positive re-evaluation	●		●	○	○	○		●	●	○			●	●	○			○
Problem-solving	●	●				○	●	●				○	○	●	○	●		●
Resilience	●		●		○	○		●	●		○			○	●		●	●
Raw coverage	0.37	0.35	0.34	0.56	0.56	0.56	0.48	0.39	0.37	0.65	0.63	0.63	0.22	0.22	0.20	0.28	0.25	0.24
Unique coverage	0.01	0.01	0.01	0.01	0.00	0.01	0.05	0.05	0.01	0.02	0.01	0.01	0.09	0.01	0.02	0.03	0.01	0.00
Consistency	0.88	0.89	0.89	0.85	0.84	0.84	0.85	0.87	0.86	0.85	0.85	0.84	0.84	0.88	0.82	0.92	0.94	0.93
Overall solution consistency			0.83			0.77			0.80			0.75			0.80			0.91
Overall solution coverage			0.59			0.78			0.65			0.88			0.31			0.53

● = presence of condition. ○ = absence of condition. Expected vector for high levels of life satisfaction, and positive and negative affect: 1.0.0.1.0.1.0 (0: absent; 1: present); expected vector for low levels of life satisfaction, and positive and negative affect: 0.1.1.0.1.0.1, using the format of (Fiss, 2011).

## Data Availability

Data from this study are available on request to the corresponding author.

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
