# Peer review of "Are Coping Strategies, Emotional Abilities, and Resilience Predictors of Well-Being? Comparison of Linear and Non-Linear Methodologies"

_ijerph, 2022, doi:10.3390/ijerph19127478_

Round 1
Reviewer 1 Report
The study is well designed and the information analysis proposal is appropriate to the objectives; however, it is necessary to clarify whether the tests adapted to Google forms were previously piloted and the results obtained.
And in the sentence "In the first place, descriptive statistics were conducted for the total sample; including" 149
"central tendency (median), dispersion (standard deviation), position (percentiles), and" 150, it apparently refers to the mean instead of the median.
Author Response
We are grateful for the reviewer's indication. This was indeed an error. The term "median" has been changed to "mean".
We did not conduct a pilot study of this work. However, the instruments used in this study have been used in other research carried out by the team, with good results of internal consistency. In addition, all respondents who did not respond to the battery completely or who responded randomly were eliminated. We have added this information to the inclusion criteria.
Reviewer 2 Report
The manuscript addresses an interesting topic on the configuration of resilience to stressful situations. The authors assess predictors of well-being with quite interesting results, although the study does not give a clear application in the context of a predictive model.
There are a couple of inconsistencies in the theoretical approach that are perceived and that is the absence of a common thread to give logic and coherence to the approach. Although I must say that the authors developed a good structure for the presentation of their study.
On the other hand, the conclusions are poor and do not express the abundant information analysed in the results; the authors need to expand and improve this section. In clear absence of a good theoretical argumentation and the literature supporting the study is poor and outdated, it needs to be substantially expanded and considered for updating.
Finally, I recommend to the authors that from their results they would be in a position to propose a public policy that shows the potential of predictive welfare models.
Author Response
Finally, I recommend the authors to use their results to propose a public policy that shows the potential of predictive welfare models.
We are grateful for the reviewer's suggestions, which have been of great value in improving the manuscript. Following his recommendations, the conclusions section has been expanded, trying to capture the relevance of the results found. The practical application of the main results obtained has been emphasised, following the reviewer's valuable suggestion. We felt it was necessary to emphasise this as he indicated.
The introductory section has also been revised, adding additional information to contribute to coherence and improve the thread of the argumentation.
Finally, some recent references have been included to reinforce the main ideas and conclusions.
Reviewer 3 Report
Comments
1. The empirical context (i.e., the focus on Spain) should be better motivated in the introduction.
2. The sample size is quite limited (N=427). This limits the analyses that can presented using the data.
3. The revised version should provide more information about the data and non-response. This is important because the analyses are based on the use of small sample size. Was non-response to the survey that is used in the analyses random or not? Low life satisfaction may lead to lower likelihood to respond to the survey data.
4. Women are significantly over-presented in the data. Why?
5. Does self-reported information contain systematic measurement error? Does this have implications for the interpretation of the results that are presented in the paper?
6. Perceived working conditions are an important determinant of (life) satisfaction (https://doi.org/10.1016/j.jebo.2012.09.005). This issue should be stated in the revised version.
7. The empirical analyses present average effects for all individuals. The paper does not consider the potential heterogeneity in the effects. The relationships can differ significantly e.g., by age. The relatively small sample size (N=427) limits the analyses.
8. What is the external validity of the results that are presented in the paper?
Author Response
- The empirical context (i.e., the focus on Spain) should be better motivated in the introduction.
We have included in the introduction the need to address and expand knowledge of the behaviour of these variables in the Spanish population.
- The sample size is quite limited (N=427). This limits the analyses that can presented using the data.
Thanks to the reviewer for this suggestion. We agree that the sample is tight, so we have reflected this in the limitations. However, it is adequate for the analyses that have been carried out. We hope to expand the sample in the future.
- The revised version should provide more information about the data and non-response. This is important because the analyses are based on the use of small sample size. Was non-response to the survey that is used in the analyses random or not? Low life satisfaction may lead to lower likelihood to respond to the survey data.
Forgive our lack of knowledge, we do not know if we have understood your suggestion correctly. The sample used was for convenience. We do not know, given the type of evaluation (online), how many people would have been eligible. All items in the battery were mandatory, so only those that were fully answered were saved in the google forms. Please let us know if we have responded to your suggestion.
- Women are significantly over-presented in the data. Why?
La sobre representación de las mujeres en la muestra es debida a la mayor participación por parte de estas en este tipo de estudios. Pese a que es una limitación que se encuentra en muchos de los trabajos de investigación en psicología, esto se ha señalado como una limitación y consideramos que debería ser un aspecto que subsanar en futuros estudios.
- Does self-reported information contain systematic measurement error? Does this have implications for the interpretation of the results that are presented in the paper?
As far as we know, there have been no systematic measurement errors. Before conducting the relevant analyses, the database has been scanned and prepared in order to detect these and other errors.
- Perceived working conditions are an important determinant of (life) satisfaction (https://doi.org/10.1016/j.jebo.2012.09.005). This issue should be stated in the revised version.
We appreciate the reviewer's suggestion and take it into consideration for future studies. Unfortunately, this variable was not assessed, so we are unable to provide results in this respect. However, we consider that this would add value to the study, so it has been reflected in future lines, making explicit reference to the work suggested by the reviewer, which has inspired us to follow this line in the future.
- The empirical analyses present average effects for all individuals. The paper does not consider the potential heterogeneity in the effects. The relationships can differ significantly e.g., by age. The relatively small sample size (N=427) limits the analyses.
We are aware that the number of participants is a limitation. Therefore, we assume it in the discussion. As well as the possibilities we would have in the future, if the sample is enlarged. As you have mentioned.
Thank you for your suggestion.
- What is the external validity of the results that are presented in the paper?
We understand your concern about the generalisability of the data. As noted in the discussion, the main limitation of the study is its small sample size and characteristics. This aspect limits the extension of the results to the general population. The results should be interpreted with caution. This is why we include this information in the discussion. Nevertheless, we have tried to employ robust statistical analyses to improve the implications and innovation of our work.
Reviewer 4 Report
The paper examines whether Emotional Intelligence and resilience are predictor of well-being. The quality of the paper is good enough to publish in the journal.
This issue is quite important for the most of the readers. However, the results are not so surprising, because it is expected that the high level of EI would be correlated with well-being. We would be happy if EI is broken down into various factors, and examine which factors are important for well-being. The importance of factors might be different among different attributes such as sex or age group. Those analysis might make it clear how EI is related with well-being.
Author Response
We appreciate the reviewer's assessment. We agree that the sociodemographic variables mentioned by the reviewer, among others, could be relevant explanatory variables. However, the aim of the paper focused on examining the role of the 3 dimensions of EI, the 3 problem-focused coping strategies and resilience because the authors wanted to expose the relevance of psychological variables amenable to training in order to promote knowledge about them for the design of future interventions to improve well-being. The analyses carried out (fsQCA) do not admit more than 7 independent variables. This is why, in the present study, no more variables were included in the analysis. Nevertheless, we consider the information pointed out by the reviewer to be very valuable and we include it as information regarding future lines of research. This will allow us to have a broader view of the relevance of socio-demographic factors and will contribute to the descriptive study of predictors of well-being.
We are grateful for his suggestion.
Round 2
Reviewer 3 Report
I happy with the paper.